# The Effect of the 14:10-Hour Time-Restricted Feeding (TRF) Regimen on Selected Markers of Glucose Homeostasis in Diet-Induced Prediabetic Male Sprague Dawley Rats

**DOI:** 10.3390/nu17020292

**Published:** 2025-01-15

**Authors:** Sthembiso Msane, Andile Khathi, Aubrey Mbulelo Sosibo

**Affiliations:** Department of Human Physiology, School of Laboratory Medicine and Medical Sciences, College of Health Sciences, University of KwaZulu-Natal, Durban 4000, South Africa; 219023719@stu.ukzn.ac.za (S.M.); sosiboa@ukzn.ac.za (A.M.S.)

**Keywords:** intermittent fasting, prediabetes, insulin signalling pathway, glucose regulation

## Abstract

Background: Prediabetes is a condition that often precedes the onset of type 2 diabetes mellitus (T2DM). Literature evidence indicates that prediabetes is reversible, making it an important therapeutic target for preventing the progression to T2DM. Several studies have investigated intermittent fasting as a possible method to manage or treat prediabetes. Objectives: This study evaluated the impact of a 14:10-hour time-restricted feeding (TRF) regimen on leptin concentration, insulin sensitivity and selected markers associated with the insulin signalling pathway and glucose homeostasis in diet-induced prediabetic rats. Methods: Twenty-four male Sprague Dawley rats were obtained and randomly divided into two dietary groups: group 1 (*n* = 6) received a standard diet and water, while group 2 (*n* = 18) was provided a high-fat, high-carbohydrate (HFHC) diet supplemented with 15% fructose for a period of 20 weeks to induce prediabetes. After confirming prediabetes, an intermittent fasting (IF) regimen was assigned to the rats while also having untreated and metformin-treated prediabetic rats serving as controls. Results: Both IF and HFHC-Met groups yield significantly lower blood glucose, leptin and BMI results compared to the prediabetic group. The IF group yielded significantly lower insulin, HOMA-IR and HbA1C than both controls. Conclusions: The study showed the potential of IF in alleviating prediabetes-induced dysregulation of glucose homeostasis and therefore warrants further investigations into its use in the management of prediabetes.

## 1. Introduction

Type 2 diabetes mellitus (T2DM) is a metabolic disorder characterized by increased blood glucose concentrations due to insulin resistance and a relative deficiency in insulin production [1]. Globally, the prevalence of diabetes is rising at an alarming rate, with nearly half a billion individuals currently affected [2]. This number is expected to grow by 25% in 2030 and 51% in 2045 [2]. Although various treatments for T2DM are available, none have been shown to completely reverse the condition. The onset of T2DM is commonly preceded by a state known as prediabetes [3].

Prediabetes is a metabolic state characterized by impaired fasting glucose (IFG) and impaired glucose tolerance (IGT) that have not reached the diagnostic criteria for T2DM [3]. Despite the World Health Organization not including HbA1c as a criterion, the American Diabetes Association (ADA), on the other hand, defines prediabetes as having an IFG level between 5.6–6.9 mmol/L, an IGT range of 7.8–11.0 mmol/L and an HbA1c level between 5.7–6.4% [4]. A recent study indicated that the global prevalence of prediabetes is projected to surpass 400 million people by 2045 [5]. Studies have shown that prediabetes is reversible and has thus become a therapeutic target to prevent the onset of T2DM. Primary strategies for managing prediabetes include increased physical activity, dietary modifications and the administration of metformin as well as intermittent fasting [6].

Intermittent fasting (IF) is a dietary regimen that involves cycling between periods of food intake and extended intervals of calorie restriction [7]. Common IF regimens encompass alternate-day fasting, the 5:2 fasting protocol and time-restricted feeding [8,9]. Time-restricted feeding (TRF) is an IF protocol with a specified time of prolonged fasting practiced by adhering to 14 hr of abstinence from food and 10 hr of food intake within 24 h [10]. The 14:10 hour IF protocol has been shown to decrease HbA1c levels, fasting blood glucose effectively and body weight (−3.15%) and enhance lipid profiles in obese T2DM patients [11,12].

IF has shown potential benefits for individuals with T2DM. These benefits include decreases in fasting insulin levels, insulin resistance, IGT, triacylglycerol levels and fasting glucose levels and improvements in insulin sensitivity [9,13,14,15,16]. However, there is limited knowledge regarding the impact of IF on glucose regulation in individuals with prediabetes and an insufficient understanding of its role within the insulin signalling pathway in the long term.

Research indicates the widespread use of animal models to study the metabolic changes associated with both T2DM and prediabetes as well as the effects of various intervention strategies to manage these conditions [17,18]. Our laboratory employs a diet-induced animal model of prediabetes which has been shown to mimic the human condition [19]. Using this model, the study sought to investigate the effects of a 14:10-hour TRF regimen on glucose regulation in prediabetic male Sprague Dawley rats.

## 2. Materials and Methods

### 2.1. Animals and Housing

All animal experimentation was approved by the Animal Research Ethics Committee (AREC) of the University of Kwa-Zulu Natal. The authors, the Biomedical Research Unit staff and the veterinarian were informed about the group assignments at various stages of the experiment. Three-week-old male Sprague-Dawley rats were bred and housed in the Biomedical Research Unit (BRU) (ethics no. AREC/00006223/2023(00022008)) of the University of Kwa-Zulu Natal were used in the study. The animals were maintained under standard laboratory conditions of constant temperature (22 ± 2 °C), carbon dioxide (CO_2_) content (<5000 p.m.), relative humidity (55 ± 5%) and illumination (12 h light/dark cycle, lights on at 07h00). The noise level was maintained at less than 65 decibels approved. The animals were allowed access to food and fluids ad libitum. The animals acclimatized to their new environment for 1 week while consuming standard rat chow and tap water before the induction of prediabetes by exposure to a well-established experimental diet (HFHC) shown in Appendix A.

### 2.2. Prediabetes Induction

Twenty-four male Sprague Dawley rats were obtained and randomly divided into two dietary groups: group 1 and group 2. Experimental prediabetes was induced in these rats following a previously established protocol [19]. Group 1 (*n* = 6) received a standard diet and water, while group 2 (*n* = 18) was provided a high-fat, high-carbohydrate (HFHC) diet supplemented with 15% fructose for a period of 20 weeks. After this period, the American Diabetes Association’s criteria for prediabetes were applied to all animals. Animals were classified as prediabetic if they exhibited fasting blood glucose levels between 5.6 and 6.9 mmol/L and 2-h glucose levels between 7.1 and 11.1 mmol/L in an oral glucose tolerance test (OGTT). Animals falling below these thresholds were considered non-prediabetic.

To calculate the sample size for this study, we utilized G power software (version 3.1), which is capable of accommodating both distribution-based and design-based modes as well as the minimum number of rats to perform statistical analysis. The calculations from Algorithm 1 determined that at least 12 rats are required to perform the statistical analysis. In alignment with ethical considerations, we opted to adhere to the minimum of 6 per group. The input and output are displayed as follows:

**Algorithm** **1:** A priori power analysis for a one-way ANOVA with fixed effects, calculating the required sample size. The analysis is based on the specified input parameters: effect size, significance level, desired power, and number of groups.F tests—ANOVA: Fixed effects, omnibus, one-wayAnalysis: A priori: Compute required sample sizeInput:    Effect size f      =   1.335799     α err prob         =  0.05     Power (1 − β err prob)       =   0.80     Number of groups       =   4Output: Non-centrality parameter λ =   21.4123076     Critical F   =     4.0661806     Nu-merator *df*       =  3     Denominator *df*    =  8     Total sample size    =  12     Actual power     =  0.8681756

### 2.3. Oral Glucose Tolerance Response

An oral glucose tolerance test was performed on all animals to assess their glucose tolerance response. This test was completed after carbohydrate loading, following a well-established laboratory technique [20]. After fasting for 12 h, glucose levels were determined at time 0. Then, a monosaccharide syrup was administered orally using an 18-gauge gavage needle that is 38 mm long and curved, with a 21/4 mm ball end (Able Scientific, Canning Vale, Australia). The glucose concentration was determined by collecting blood using the tail-prick method [21] and measuring glucose concentrations using a OneTouch select glucometer (Lifescan, Mosta, Malta, UK). Glucose levels were subsequently assessed at 15-, 30-, 60- and 120 min following carbohydrate loading.

### 2.4. Intermittent Fasting Protocol

After measuring glucose levels, an intermittent fasting (IF) protocol was introduced to prediabetic male Sprague Dawley rats. As shown in Figure 1, group 2 was split into three subgroups: A, B and C. Group A (*n* = 6) continued with the HFHC diet supplemented with 15% fructose for the entire experimental period (12 weeks). Group B (*n* = 6) followed a well-established IF protocol [11], where food was removed at 17:00 in the evening and returned at 07:00 the next morning, for 5 days each week over a 12-week period. During the eating window, the Sprague Dawley rats were provided with an HFHC diet and drinking water with 15% fructose. However, during the fasting period, they were only given water. Group C (*n* = 6) was introduced to metformin. This group continued with HFHC diet supplemented with 15% fructose and was treated with an oral dose of metformin (500 mg/kg, Sigma-Aldrich, St. Louis, MO, USA every third day for 12 weeks. Every 4 weeks, body weight, calorie intake, oral glucose response and fasting glucose were measured for all groups. Furthermore, body mass index was calculated using the following formula:

BMI = weight (g)/height (cm^2^).

**Figure 1 nutrients-17-00292-f001:**
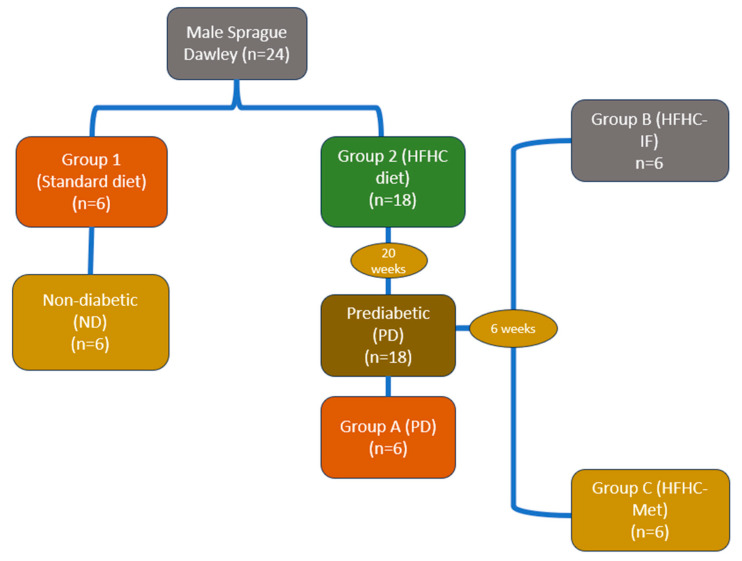
Experimental protocol showing prediabetes induction, intermittent fasting and HFHC-Met implementation on male Sprague Dawley rats. ADA criteria were used to identify prediabetes.

### 2.5. Blood Collection and Harvesting

All animals were anaesthetized using Isofor (100 mg/kg) (Safeline Pharmaceuticals (Pty) Ltd., Roodepoort, South Africa) in a gas anaesthetic chamber (Biomedical Resource Unit, University of KwaZulu-Natal, Durban, South Africa), with a 3-min inhalation period. Blood was drawn via cardiac puncture and placed into individual pre-cooled heparinized containers. The blood samples were then centrifuged to separate the plasma (Eppendorf centrifuge 5403, Wesseling-Berzdorf, Germany) at 4 °C, 503× *g* for 15 min. Following this, the skeletal muscle and the liver were snap-frozen in liquid nitrogen before storage in a BioUltra freezer (Snijers Scientific, Tilburg, The Netherlands) at −80 °C until biochemical analysis. 

### 2.6. Biochemical Analysis

Plasma insulin, leptin and glycated haemoglobin were measured using their respective sandwich and rat-competitive ELISA kits (Elabscience Biotechnology Co., Ltd., Wuhan, China) according to the manufacturer’s instructions. Five wells were prepared for standard points and one well for the blank. To each well, 50 μL of standard dilutions, blank and samples, was added, followed by 50 μL of detection reagent A to each well immediately. The plate was gently shaken and covered with a plate sealer before incubating for 1 h at 37 °C. Detection reagent A, which may appear cloudy, was warmed to room temperature and mixed until uniform. The solution was aspirated, and each well was washed with 350 μL of 1× wash solution, repeated 3 times. After washing, 100 μL of detection reagent B working solution was added to each well and incubated for 30 min at 37 °C and then washed 5 times as before. Subsequently, 90 μL of substrate solution was added and incubated for 10–20 min at 37 °C, protected from light. The liquid turned blue, and 50 μL of stop solution was added, turning the liquid yellow. The plate was tapped gently to mix, ensuring no bubbles or fingerprints, and the microplate reader was used for measurement at 450 nm.

### 2.7. Expression of Insulin Signalling Pathway Receptors via Real-Time-PCR

The liver and skeletal muscle tissues were collected and homogenized, and total RNA was isolated using a ReliaPrep simplyRNA Tissue Miniprep System (Promega, Madison, WI, USA) according to the manufacturer’s protocol. A Nanodrop 2000 (Thermo Scientific, Roche, South Africa) was used to determine the purity and concentration of RNA. A purity ratio (A260/A280) of 1.7–2.1 was considered acceptable for conversion of RNA to cDNA. Following the manufacturer’s instructions (GoTaq^®^ 2-Step RT-qPCR System as a cDNA synthesis kit, Promega, USA). Total RNA was reverse-transcribed to cDNA. To perform the PCR amplification on the Applied Biosystem Quantstudio^®^5 RT PCR system (Thermo-Scientific, Applied Biosystem Quantstudio^®^5, Centurion, South Africa), the Go Taq^®^ qPCR Master Mix was used. The primer sequences (Metabion, Planegg, Germany) used in this study are listed in Table 1 below. PCR was performed using the following cycling conditions: an initial Go Taq DNA Polymerase activation (1 cycle at 95 °C for 2 min), a denaturation (40 cycles at 95 °C for 15 s) and annealing and extension (40 cycles at 60 °C for 1 min). Melting curve analysis was performed at 95 °C for 30 s, 65 °C for 20 s and 95 °C at a ramp rate of 0.05 °C/s and a continuous fluorescence measurement, followed by a final cooling step at 40 °C for 60 s. The RT-qPCR results were analysed using the 2^−ΔΔCq^ comparative method relative to the control groups [22]. The housekeeping gene used in this study was glyceraldehyde-3-phosphate dehydrogenase (GAPDH).

### 2.8. Glycogen Assay

Glycogen levels in the liver and skeletal muscle tissues were analysed using a standard laboratory protocol [23,24]. Muscle and liver samples were weighed and heated with 2 mL of 30% potassium hydroxide (KOH) at 100 °C for 30 min separately. To halt the reaction, 0.194 mL of 10% disodium sulphite was immediately added to the mixture. After cooling, glycogen precipitate formed. A 200 µL aliquot of the cooled mixture was mixed with 200 µL of 95% ethanol. The glycogen precipitate was pelleted, washed and dissolved in 1 mL of water. Subsequently, 4 mL of anthrone solution (0.5 g in 250 mL sulphuric acid) was added, and the mixture was boiled for 10 min. After cooling, absorbance was measured at 620 nm using the Spectrostar Nano spectrophotometer (BMG Labtech, Ortenburg, Germany). Glycogen concentrations were determined using a glycogen standard curve.

### 2.9. Statistical Analysis

All data are expressed as means ± standard error of means (SEM). GraphPad Prism Instant Software (version 8.00, GraphPad Software, San Diego, CA, USA) was used for statistical analysis. All terminal data were analysed using the normality and lognormality test and a one-way ANOVA test to assess differences between control and experimental groups. Values of *p* < 0.05 were considered statistically significantly different between the compared groups.

## 3. Results

### 3.1. Calorie Intake

The 24 h calorie intake was calculated in all experimental groups during the 12-week treatment period. The results (Table 2) showed significantly (*p* < 0.05) higher calorie intake in the PD control group in comparison to the NPD group. However, the 14:10 TRF-treated group showed significantly (*p* < 0.05) lower calorie intake in weeks 0, 4, 8 and 12 in comparison to the PD control. The HFHC-Met group similarly exhibited significantly (*p* < 0.05) lower calorie intake in weeks 0, 4, 8 and 12 compared to the PD control.

### 3.2. Bodyweight Change

The average body weight and the percentage changes in body weight were measured and calculated during the 12-week treatment period. The results (Table 3) showed significantly (*p* < 0.05) higher body weight in the PD control group in comparison to the NPD group. However, the 14:10 TRF-treated group showed significantly (*p* < 0.05) lower body weights in comparison to the PD control. The HFHC-Met group exhibited significantly (*p* < 0.05) similar results compared to the PD control.

### 3.3. Body Mass Index

The body mass index was calculated in all experimental groups during the 12-week treatment period. The results (Table 4) showed significantly (*p* < 0.05) higher body mass index in the PD control group in comparison to the NPD group. However, the 14:10 TRF-treated group showed significantly (*p* < 0.05) lower body mass index in comparison to the PD control. The HFHC-Met group exhibited significantly (*p* < 0.05) improved body mass index results compared to the PD control.

### 3.4. 2-h Oral Glucose Tolerance Tests

The 2-h oral glucose tolerance tests (OGTT) were measured in all experimental groups during weeks 0, 4, 8 and 12 of the treatment periods. The results (Table 5, Figure 2) showed significantly (*p* < 0.05) higher OGTT in the PD control group in comparison to the NPD group. However, in week 12 of treatment, the 14:10 TRF-treated group showed significantly (*p* < 0.05) improved glucose tolerance in comparison to the PD control. The HFHC-Met group also exhibited significantly (*p* < 0.05) improved glucose tolerance results compared to the PD control.

### 3.5. Fasting Glucose, Plasma Insulin, HOMA-IR and Blood HbA1c

The fasting glucose (FG), insulin levels, homeostatic insulin resistance and glycated haemoglobin were measured in all experimental groups during the 12-week treatment period. The results (Table 6) showed significantly (*p* < 0.05) higher FG and HbA1c in the PD control group in comparison to the NPD group. However, in week 12 of treatment, the 14:10 TRF-treated group showed significantly (*p* < 0.05) improved insulin concentration, HOMA-IR and HbA1c in comparison to the PD control. The HFHC-Met group also exhibited significantly (*p* < 0.05) improved HbA1c results compared to the PD control.

### 3.6. Plasma Leptin

The plasma leptin concentrations were measured in all experimental groups during the 12-week treatment period. The results (Figure 3) showed significantly (*p* < 0.05) higher leptin concentration in the PD control group in comparison to the NPD group. However, the 14:10 TRF-treated group showed significantly (*p* < 0.05) improved leptin concentration in comparison to the PD control. The HFHC-Met group also exhibited significantly (*p* < 0.05) improved leptin concentration results compared to the PD control.

### 3.7. Expression of IRS1, IRS2, Akt, PI3K, mTORC1 and GLUT 4

The expression of IRS1, IRS2, Akt, PI3K, mTORC1 and GLUT 4 were measured in all experimental groups during the 12-week treatment period. The results (Figure 4) showed significantly (*p* < 0.05) higher IRS1, IRS2, Akt, PI3K and mTORC1 in the PD control group in comparison to the NPD group. However, the 14:10 TRF-treated group showed significantly (*p* < 0.05) improved IRS1, IRS2, Akt, PI3K, mTORC1 and GLUT4 in comparison to the PD control. The HFHC-Met group also exhibited significantly (*p* < 0.05) improved IRS1, Akt, PI3K, mTORC1 and GLUT4 results compared to the PD control. Interestingly, the HFHC-Met group also demonstrated significantly higher IRS2 compared to the PD control group.

### 3.8. Glycogen Concentration in Liver and Skeletal Muscle

The glycogen concentrations were measured in the liver and skeletal muscles in all experimental groups during the 12-week treatment period. The results (Figure 5) showed significantly (*p* < 0.05) higher liver glycogen concentration in the PD control group in comparison to the NPD group. Further, the 14:10 TRF-treated group showed substantially lower liver glycogen concentration in comparison to the PD control. Furthermore, the HFHC-Met group exhibited significantly (*p* < 0.05) lower glycogen concentration results compared to the PD control. Interestingly, the opposite was reported in skeletal muscle glycogen concentration.

## 4. Discussion

Type 2 diabetes mellitus (T2DM) remains a major factor contributing to global mortality and morbidity rates [2]. T2DM is a chronic hyperglycaemic condition triggered by a preceding loss of β-cell insulin secretion and insulin resistance [1]. The onset of T2DM is often preceded by prediabetes. Prediabetes can be characterized by elevated blood glucose levels that do not meet the diagnostic criteria for diabetes mellitus [25]. Despite the World Health Organization not including HbA1c as a criterion, the American Diabetes Association (ADA), on the other hand, defines prediabetes as having an impaired fasting glucose (IFG) level between 5.6–6.9 mmol/L, an impaired glucose tolerance (IGT) range of 7.8–11.0 mmol/L and an HbA1c level between 5.7–6.4% [4]. Research indicates that prediabetes is reversible, making it a key focus for preventing the progression of T2DM. Primary strategies for managing prediabetes include enhancing physical activity, adopting dietary changes and using metformin [6]. Despite the availability of these management strategies, prediabetes remains a significant challenge. Recent studies have indicated that intermittent fasting has shown promising effects on the management of T2DM. Intermittent fasting refers to various dietary approaches where individuals alternate extended periods of normal calorie consumption with significantly reduced or no calorie intake intervals [26,27]. The timing of fasting and feeding periods varies among different IF protocols, such as the 5:2 diet, alternate-day fasting and time-restricted feeding [12,26,28]. The 14:10 hour IF protocol has been shown to decrease HbA1c levels, fasting blood glucose effectively and body weight and enhance lipid profiles in obese T2DM patients [11,12]. However, the impact of intermittent fasting on glucose regulation in individuals with prediabetes remains poorly understood, and its long-term role in the insulin signalling pathway is not yet well-defined. This study utilized a diet-induced animal model of prediabetes, known to replicate the human condition, to examine the impact of a 14:10-hour time-restricted feeding (TRF) regimen on glucose regulation in prediabetic male Sprague Dawley rats.

This study specifically investigated the effects of a 14:10-hour time-restricted feeding regimen on leptin levels, insulin sensitivity and various markers related to the insulin signalling pathway, including IRS1, IRS2, Akt, PI3K, mTORC1, GLUT4 and glucose homeostasis in diet-induced prediabetic rats. Significant changes were observed in parameters such as calorie intake, body weight, leptin concentrations, OGTT, HbA1c, insulin levels, HOMA-IR, IRS1, IRS2, Akt, PI3K, mTORC1, GLUT4 and glycogen concentration compared to the untreated prediabetic group. However, no significant differences were found in fasting glucose, OGTT at weeks 0 and 8.

Under normal conditions, body weight reflects a balance between energy intake and expenditure [29,30]. When fat stores are adequate, leptin signals the body to increase energy expenditure and reduce energy intake [31,32]. However, an energy imbalance occurs in the prediabetic state, leading to increased body weight, body mass index (BMI) and leptin resistance [31,32,33]. Our findings align with existing literature, showing significantly higher body weight, BMI and leptin concentrations in the prediabetic group than in the non-prediabetic group throughout the experimental period, despite a minimal body weight percental change of ±0.3%. Studies suggest that the prediabetic group experiences higher energy intake relative to expenditure due to increased leptin levels [34]. Interestingly, the intermittent fasting (IF) group exhibited significantly lower body weight, BMI and leptin levels than the prediabetic group. This suggests that fasting periods may help restore energy balance by activating leptin sensitivity, promoting reduced caloric intake [35,36]. A study revealed that the IF can activate the insulin signalling pathway regardless of unchanged body weight [37]. Additionally, the high-fat, high-carbohydrate, metformin (HFHC-Met) group showed notably lower body weight, BMI and leptin concentrations than the prediabetic group. Growing evidence indicates that metformin-induced weight loss is driven by its effects on hypothalamic appetite-regulation centres including increasing leptin sensitivity, changes in the gut microbiome and the reversal of aging-related factors [38]. We propose that the body weight and BMI changes observed in both the IF and HFHC-Met groups were driven by leptin sensitivity, which further contributed to reduced calorie intake.

Impaired glucose tolerance (IGT), along with impaired fasting glucose (IFG) and elevated glycated haemoglobin (HbA1c), are recognized as contributing factors to moderate insulin resistance in insulin-dependent tissues [39]. Under normal conditions, an increase in blood glucose levels following meals prompts insulin release, which promotes glycogenesis [40]. However, in the prediabetic state, insulin fails to regulate blood glucose, leading to impaired blood glucose and glucose intolerance [41]. Our findings are consistent with prior research, showing significantly higher blood glucose levels in the prediabetic group than in the non-prediabetic group throughout the experimental period. Interestingly, after one week of the IF regimen intervention, the 2-h oral glucose tolerance test showed significantly lower results in the IF group compared to the prediabetic group. In contrast, fasting glucose levels showed no significant difference between the IF and prediabetic groups. However, HbA1c indicated significantly lower glycaemia in the IF group compared to the prediabetic group. This suggests that the IF regimen effectively regulates glycaemic metabolism by activating the insulin signalling pathway, thereby promoting glucose uptake through GLUT4 and reducing liver glucose release [37,42]. A study demonstrated that improvements in glucose metabolism under the IF regimen were mainly attributed to the suppression of hepatic glucose output, mediated by the degradation of the liver glucocorticoid receptor and its downstream transcription factor, Kruppel-like factor 9 [42]. Additionally, the HFHC-Met group showed improved 2-h oral glucose tolerance and HbA1c results compared to the prediabetic group. Metformin has been shown to regulate glucose metabolism by inhibiting gluconeogenesis. Additionally, it activates AMPK, which improves insulin sensitivity by modulating fat metabolism and promotes the translocation and expression of GLUT4 [8,43,44].

Insulin plays a critical role in stabilizing blood glucose levels. In the normal state, insulin is released in response to elevated blood glucose levels to regulate glucose concentrations [45,46]. However, in the prediabetic state, elevated glucose levels trigger the release of large amounts of insulin, eventually causing cells to become resistant to its effects [47,48]. Our findings align with these studies, revealing significantly higher insulin concentration in the prediabetic group compared to the non-prediabetic group. This indicates the presence of insulin insensitivity. Studies have demonstrated that the presence of insulin insensitivity leads to disruption of the insulin signalling pathway and the failure of GLUT4 translocation to cells, which hinders glucose uptake [49,50]. Interestingly, at the end of the 12-week experimental period, the IF group demonstrated significantly lower insulin concentration and HOMA-IR results than the prediabetic group. This indicates that intermittent fasting aids in controlling hyperglycaemia by maintaining insulin sensitivity. Research has demonstrated that IF decreases the body’s energy intake, resulting in reduced insulin production and elevated AMPK levels, which may contribute to improved insulin sensitivity and glucose homeostasis [37,41,51,52]. This has been shown to occur in the fasted state, enabling the body to utilize and manage blood glucose. Furthermore, in this study, the HFHC-Met group exhibited no significant differences in insulin levels and HOMA-IR compared to the prediabetic control group. This may be because the metformin-treated group remained on the HFHC diet throughout the study. It is recommended that to obtain maximal efficacy from metformin, it must be combined with lifestyle modifications such as dietary intervention and increased physical activity, which may enhance its efficacy [53,54]. However, there is reported poor patient compliance as patients tend to have an over-reliance on metformin and neglect lifestyle modifications. This phenomenon was also observed in this study. Moreover, another study demonstrated that the effectiveness of metformin is dependent on the dosage [55].

Insulin binds to its receptor on the surface of insulin-responsive cells, triggering the receptor’s activation. The activated insulin receptor then recruits insulin receptor substrate 1 (IRS1) and IRS2, which are phosphorylated at specific tyrosine sites [56,57]. These phosphorylated IRS proteins act as docking platforms for downstream signalling molecules. One such molecule, PI3K, binds to the phosphorylated IRS1/2 and becomes activated. PI3K then converts phosphatidylinositol-4,5-bisphosphate (PIP2) into phosphatidylinositol-3,4,5-trisphosphate (PIP3) at the cell membrane [58,59]. PIP3 facilitates the activation of Akt (protein kinase B) by recruiting it and enabling phosphorylation by phosphoinositide-dependent kinase-1 (PDK1). Akt plays a pivotal role in regulating insulin’s metabolic functions [59]. It activates the mechanistic target of rapamycin complex 1 (mTORC1) by inhibiting the tuberous sclerosis complexes (TSC1/TSC2), which normally restrains mTORC1 [59,60]. The activation of mTORC1 supports protein synthesis, cell growth and glucose metabolism. Additionally, Akt promotes the movement of glucose transporter 4 (GLUT 4) vesicles to the cell membrane, enabling glucose uptake into the cell [61,62]. In the prediabetic state, cells develop a reduced sensitivity to insulin due to impairments in IRS1/2 signalling and associated downstream pathways [50]. To compensate, the pancreas increases insulin secretion, which intensifies feedback inhibition and places additional strain on β-cells [63,64]. Interestingly, unlike the prediabetic state, intermittent fasting (IF) effectively restored the insulin signalling pathway by regulating IRS1/2, Akt, PI3K and mTORC1 while promoting GLUT4 activity. Similarly, studies have shown that the IF regimen can effectively restore the insulin signalling pathway [37,41,51,60,62,65,66,67,68]. Additionally, this study revealed a significant decrease in IRS1, Akt, PI3K and mTORC1 and a significantly increased GLUT 4 activity. This is in line with a study demonstrating that metformin in the presence of insulin directly stimulates IRS-1 and-2 to activate Akt via PI3K [69].

Glycogen levels in the liver and skeletal muscle were evaluated throughout the 12-week experimental period. Under normal physiological conditions, insulin enhances glucose uptake in the liver and activates glycogen synthase, promoting glycogen storage [70,71]. In the liver, glucose transport is mediated by GLUT 2, a bidirectional transporter that facilitates glucose movement in and out of cells based on the concentration gradient [72]. In conditions such as insulin resistance or hyperglycaemia, GLUT 2 facilitates an increased influx of glucose into hepatocytes [73,74]. Since GLUT 2 operates along the concentration gradient, elevated blood glucose levels drive higher glucose uptake into the liver [73]. In this study, liver glycogen levels were significantly higher in the prediabetic group compared to the non-prediabetic group, likely due to compensatory mechanisms in response to insulin resistance. Notably, the IF group displayed a marked reduction in glycogen levels compared to the prediabetic group, aligning with findings from other studies highlighting the effectiveness of an IF regimen in regulating glucose storage [75,76]. Similarly, the HFHC-Met group showed results comparable to those of the prediabetic group.

In contrast, skeletal muscle glycogen levels were significantly lower in the prediabetic group compared to the non-prediabetic group, likely reflecting impaired insulin-mediated glucose uptake associated with insulin resistance [77,78]. However, the IF group exhibited a notable recovery in muscle glycogen levels, suggesting improved insulin sensitivity and glucose uptake, consistent with evidence supporting the metabolic benefits of intermittent fasting [79,80,81]. Additionally, the HFHC-Met group also demonstrated significantly increased skeletal muscle glycogen levels. The glycogen findings are in line with data on body weight, leptin, OGTT, insulin, HOMA-IR, IRS1, IRS2, PI3K, Akt, mTORC1 and GLUT 4 across all groups.

## 5. Limitations and Future Studies

Although the study demonstrated improvements in markers related to the insulin signalling pathway, a significant limitation was the measurement of proteins in actual tissue to validate whether the gene expression changes identified by PCR correspond to protein-level alterations. Due to limited funding at this time, these could not be conducted, but these will be addressed in future studies by using Western blotting and immunohistochemistry to support the PCR findings. This study showed significant effects on glycogen formation in the livers and skeletal muscle of the 14:10 TRF-treated diet-induced prediabetic rats. This suggests that this intervention may have effects on the overall physiology of these tissues therefore future studies need to investigate the changes in the markers associated with the physiology of these tissues. Furthermore, future studies could also look at other organ systems such as the cardiovascular and renal systems as these are greatly affected by changes in glucose homeostasis.

## 6. Conclusions

In this study, the results showed that chronic ingestion of a high-carbohydrate, high-fat diet results in prediabetes. As expected, the untreated prediabetic group exhibited glucose dysregulation. However, despite being maintained on an HFHC diet, introducing an intermittent fasting (IF) regimen effectively mitigated glucose-related dysregulation. The benefits of IF were highlighted by significant improvements in GLUT4 activity, accompanied by reductions in body weight, leptin levels, HOMA-IR and insulin. These changes aid in enhancing insulin sensitivity, ensuring the proper function of IRS1 and IRS2. These proteins then effectively transmitted signals to PI3K, Akt and mTORC1, facilitating GLUT4 translocation and promoting glucose uptake into cells. Taken together, the study showed the potential of IF in alleviating prediabetes-related dysregulation of glucose homeostasis and therefore warrants further investigations into its use in the management of prediabetes. Although the diet-induced animal model of prediabetes has been shown to closely replicate the human condition, we cannot conclude that the effects of IF would be identical in humans, as factors like genetic makeup and dietary habits may influence the outcomes. Additionally, we recommend conducting further research to explore its application in humans.

## Figures and Tables

**Figure 2 nutrients-17-00292-f002:**
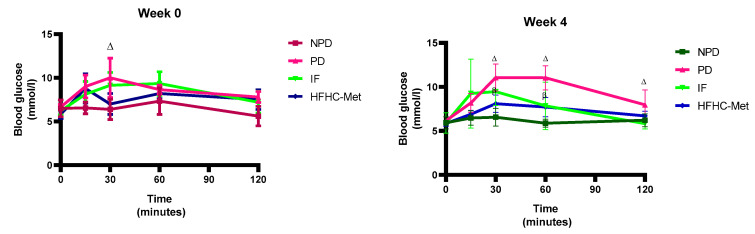
Effect of the 14:10-hour TRF regimen on the 2-h oral glucose tolerance test in prediabetic rats during the 12-week experiment. ∆ = *p* value < 0.05 denotes comparison with NPD and β = *p* value < 0.05 denotes a comparison with PD.

**Figure 3 nutrients-17-00292-f003:**
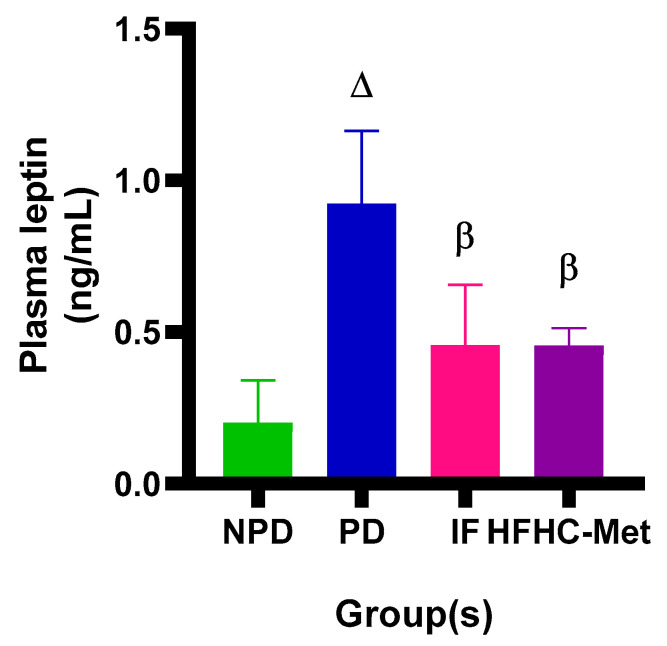
Effect of the 14:10-hour TRF regimen on the plasma leptin concentration in prediabetic rats at week 12 of the experiment. Data are presented as mean values. ∆ = *p* value < 0.05 denotes comparison with NPD and β = *p* value < 0.05 denotes a comparison with PD.

**Figure 4 nutrients-17-00292-f004:**
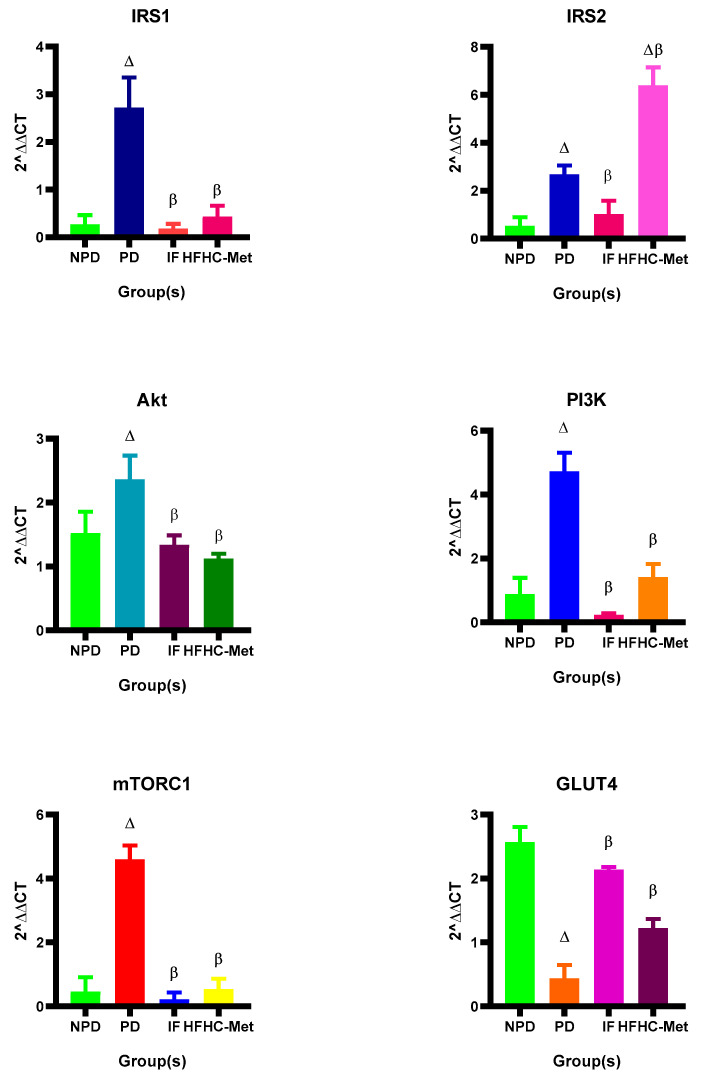
Effect of the 14:10-hour TRF regimen on the IRS1, IRS2, Akt, PI3K, mTORC1 and GLUT 4 in prediabetic rats at week 12 of the experiment. ∆ = *p* value < 0.05 denotes comparison with NPD and β = *p* value < 0.05 denotes a comparison with PD.

**Figure 5 nutrients-17-00292-f005:**
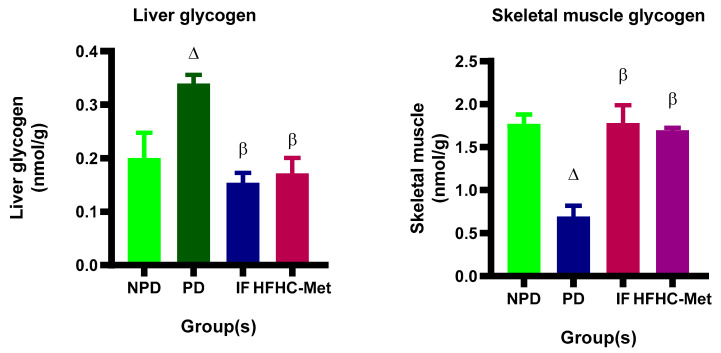
Effect of the 14:10-hour TRF regimen on the glycogen concentration on liver and skeletal tissues in prediabetic rats at week 12 of the experiment. ∆= *p* value < 0.05 denotes comparison with NPD and β = *p* value < 0.05 denotes a comparison with PD.

**Table 1 nutrients-17-00292-t001:** List of primers used in the study.

Sequence Name	Sequence
GLUT 4 F	ATCAACGCCCCACAGAAAGT
GLUT 4 R	CCTGCCTACCCAGCCAAGT
Akt F	ATGGACTTCCGGTCAGGTTCA
Akt R	GCCCTTGCCCAGTAGCTTCA
MTORC1 F	TGCAGCCTGACCAATGATGTG
MTORC1 R	CTTGTGTCCGGCAGCATCATC
IRS1 F	GCCAATCTTCATCCAGTTGC
IRS1 R	CATCGTGAAGAAGGCATAGG
IRS2 F	CTACCCACTGAGCCCAAGAG
IRS2R	CCAGGGATGAAGCAGGACTA
PI3K F	ACTGAGATGGAGACACGGAAC
PI3K R	GCATCCAAGGGTCCAGTTAGTG
GAPDH F	AGTGCCAGCCTCGTCTCATA
GAPDH R	GATGGTGATGGGTTTCCCGT

**Table 2 nutrients-17-00292-t002:** Effects of the 14:10-hour TRF regimen on 24hr calorie intake in non-prediabetic and prediabetic rats during the 12-week experimental period. ∆ = *p* value < 0.05 denotes comparison with NPD and β = *p* value < 0.05 denotes a comparison with PD. The data is represented as means ± SEM.

Groups	Week 0	Week 4		Week 8		Week 12	
	Caloric Intake (kcal/g)	Caloric Intake (kcal/g)	% Change	Caloric Intake (kcal/g)	% Change	Caloric Intake (kcal/g)	% Change
NPD	55.1 ± 2.8	58.2 ± 6.5	+5.6	61.3 ± 5.9	+5.3	65.7 ± 5.9	+7.2
PD	98.4 ± 6.9 ^∆^	110 ± 10.6 ^∆^	+11.8	112.3 ± 10.1 ^∆^	+2.1	121.8 ± 6.9 ^∆^	+8.5
IF	91.3 ± 3.3 ^β^	91.4 ± 6.6 ^β^	+0.1	93.2 ± 12.5 ^β^	+2.0	108.4 ± 12.0 ^β^	+16.3
HFHC-Met	93.2 ± 12.5 ^β^	98.9 ± 6.9 ^β^	+6.1	102.6 ± 11.7 ^β^	+3.7	112.1 ± 6.7 ^β^	+9.3

**Table 3 nutrients-17-00292-t003:** Effect of the 14:10-hour TRF regimen on the average body weight and the percentage changes in body weight in prediabetic rats during the 12-week experiment. ∆ = *p* value < 0.05 denotes comparison with NPD and β = *p* value < 0.05 denotes a comparison with PD. The data is represented as means± SEM.

Groups	Week 0	Week 4		Week 8		Week 12	
	Body Weight (g)	Body Weight (g)	% Change	Body Weight (g)	% Change	Body Weight (g)	% Change
NPD	401 ± 10.5	425 ± 12.6	+23	489 ± 17.0	+27	503 ± 16.7	+28
PD	603 ± 47.7 ^∆^	759 ± 82.8 ^∆^	+26	786 ± 97.0 ^∆^	+26	831 ± 102.5 ^∆^	+28
IF	594 ± 7.8 ^∆^	613 ± 7.5 ^β^	+25	632 ± 7.2 ^β^	+25	658 ± 7.3 ^β^	+26
HFHC-Met	542 ± 10.4 ^∆^	525 ± 21.8 ^β^	+23	570 ± 29.4 ^β^	+25	608 ± 29.0 ^β^	+27

**Table 4 nutrients-17-00292-t004:** Effect of the 14:10-hour TRF regimen on the body mass index (BMI) in prediabetic rats during the 12-week experiment. The values are expressed as mean ± SEM. ∆ = *p* value < 0.05 denotes comparison with NPD and β = *p* value < 0.05 denotes a comparison with PD. Values are depicted as means ± SEM.

	BMI (g/cm^2^)	BMI (g/cm^2^)	BMI (g/cm^2^)	BMI (g/cm^2^)
Groups	Week 0	Week 4	Week 8	Week 12
NPD	4.3 ± 0.2	4.4 ± 0.1	4.9 ± 0.1	5.2 ± 1.5
PD	4.8 ± 0.1	6.8 ± 0.4 ^∆^	7.0 ± 0.4 ^∆^	7.7 ± 0.4 ^∆^
IF	4.7 ± 0.3	4.8 ± 0.2 ^β^	5.1 ± 0.2 ^β^	6.0 ± 0.12 ^β^
HFHC-Met	4.6 ± 0.4	4.7 ± 0.1 ^β^	4.8 ± 0.2 ^β^	5.3 ± 0.09 ^β^

**Table 5 nutrients-17-00292-t005:** Effect of the 14:10-hour TRF regimen on the 2-h oral glucose tolerance test in prediabetic rats during the 12-week experiment. The data are represented as areas under the curve (AUC). ∆ = *p* value < 0.05 denotes comparison with NPD and β = *p* value < 0.05 denotes a comparison with PD. Values are depicted as means ± SEM.

Groups	Week 0	Week 4	Week 8	Week 12
NPD	789.5 ± 65.0	738.8 ± 33.6	804.3 ± 66.4	763.4 ± 76.7
PD	1033 ± 47.8 ^∆^	1152 ± 74.6 ^∆^	952.9 ± 55.5 ^∆^	972.8 ± 61.0 ^∆^
IF	1011 ± 66.0 ^∆^	923.1 ± 104.7 ^β^	988.7 ± 43.0 ^∆^	807.2 ± 53.5 ^β^
HFHC-Met	927.3 ± 47.9 ^∆^	877.0 ± 43.8 ^β^	873.8 ± 37.3 ^β^	849.8 ± 29.6 ^β^

**Table 6 nutrients-17-00292-t006:** Effect of the 14:10-hour TRF regimen on the fasting glucose (FG), insulin levels, homeostatic insulin resistance and glycated haemoglobin in prediabetic rats at week 12 of the experimental period. The values are expressed as mean ± SEM. ∆ = *p* value < 0.05 denotes comparison with NPD and β = *p* value < 0.05 denotes a comparison with PD.

Groups	Fasting Glucose (mmol/L)	Insulin (μU/mL)	HOMA IR	HbA1c (%)
NPD	5.4 ± 0.2	1.1 ± 0.2	0.3 ± 0.05	5.5 ± 0.6
PD	6.4 ± 0.1	1.9 ± 0.5	0.5 ± 0.1	8.2 ± 1.1 ^∆^
IF	6.0 ± 0.3	0.8 ± 0.2 ^β^	0.2 ± 0.05 ^β^	6.0 ± 0.5 ^β^
HFHC-Met	5.9 ± 0.4	1.8 ± 0.5	0.5 ± 0.1	6.7 ± 0.6 ^β^

## Data Availability

The original contributions presented in the study are included in the article and Appendix A, further inquiries can be directed to the corresponding author.

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
