# Peer review of "The Effect of the 14:10-Hour Time-Restricted Feeding (TRF) Regimen on Selected Markers of Glucose Homeostasis in Diet-Induced Prediabetic Male Sprague Dawley Rats"

_nutrients, 2025, doi:10.3390/nu17020292_

Round 1

Reviewer 1 Report

Comments and Suggestions for Authors

My main comments on the reviewed article Nutrients-3360025 are as follows:

1)     A clearly defined research objective is missing in the abstract section.

2)     In section 3. Results, the authors did not comment at all on the results obtained. This should be completed.

3)     The titles of tables and graphs should be corrected to follow the pattern "Effect of the 14:10-hour TRF regimen on (parameter/s) in prediabetic rats during the 12-week experiment".

The detailed comments are as follows:

1)     Page 1 line 35: Page 1, line 35: Indicate the names of the abbreviations IFG and IGT used.

2)     Page 9, line 242: “Body weight represents the total mass of the body “. This sentence is unnecessary because it contains an obvious fact.

3)     Page 9 lines 255 and 258: “Body mass index (BMI) is commonly used as an indicator of insulin resistance”. This statement is incorrect. BMI is an indicator of nutritional status, not insulin resistance. However, a high BMI, which is characteristic of obese people, predisposes them to insulin resistance. This should be corrected. A significantly higher BMI does not necessarily indicate the occurrence of insulin resistance.

4)     Page 9 line 265: “The BMI results align with the body weight findings across the groups’. This statement is unnecessary. It is difficult for the BMI index not to correspond to body mass, since it is calculated based on it.

5)     Page 10 line 309: The measurement of glycated haemoglobin (HbA1c) in the blood is not a method but an indicator used to assess carbohydrate metabolism.

6)     Page 11, lines  362-364: Note similar to that in point 4. Since HOMA-IR is calculated on the basis of fasting blood glucose and fasting insulin levels. Therefore, there must be a relationship between these parameters. This sentence should be corrected.

Reviewer 2 Report

Comments and Suggestions for Authors

The manuscript presents an intriguing study on intermittent fasting in SD rats. While the authors' exploration is indeed an interesting area of research, fundamental errors in the manuscript's structure and the basic analysis of insulin signaling are major obstacles.

Major Comments

Given that this study aims to investigate the effects of intermittent fasting and on prediabetes and/or body weight, it is essential to provide detailed information on food intake. The lack of data on food consumption makes it difficult to accurately assess the experimental results.

L140&Discussion

While mRNA expression data can provide valuable insights, it is insufficient to fully understand the impact of intermittent fasting on insulin signaling. To strengthen the conclusions, the authors should consider additional analyses, such as Western blotting to assess protein expression and phosphorylation levels, as well as activity assay of those proteins. The Discussion section lacks sufficient citations, particularly those related to the evaluation of mRNA expression. Incorporating more literature on this topic would strengthen the interpretation of the findings and provide a broader context for this research. Anyway, without evaluation of phosphorylation levels and protein abundance, the conclusions drawn in this study are difficult to accept.

Despite the significant limitations identified, the lack of discussion on these limitations in the "Limitations" section is a concern.

L167

The tables and figures in the Results section lack corresponding descriptions in the main text. The presence of legends alone does not meet the standard requirements for a well-structured scientific paper.

Minor Comments

L85

The authors state that they induced pre-diabetes "by exposure to a well-established experimental diet (HFHC)". Diet composition is critical in the experimental models, please provide detailed information about both the experimental and control diets. Specifically, please provide the exact composition of the HFHC diet, including the proportions of macronutrients (carbohydrates, fats, and proteins), as well as any specific additives or micronutrients. If a commercially available diet was used, please specify the manufacturer and catalog number.

L87

The authors state that animals were classified as pre-diabetic based on specific fasting and 2-hour post-OGTT glucose levels. However, these thresholds differ from the commonly accepted diagnostic criteria for prediabetes in humans. The authors should either revise these criteria to align with the human definition or provide a clear rationale for using the chosen thresholds.

L114

The calculated BMI for rats weighing 400g or more suggests an unusually short body length of approximately 10 cm. This seems inconsistent with the typical body length of SD rats. The authors should verify their measurements or provide a detailed explanation for the observed BMI values. Additionally, a comparison with previous studies could help clarify whether these results are within the expected range.

L216

The content presented in the first paragraph of the Discussion section

The Discussion section should focus on interpreting the results obtained in the study and relating them to the existing literature. The motivation for the research, which is more suitable for the Introduction, should be moved to its appropriate section.

Round 2

Reviewer 2 Report

Comments and Suggestions for Authors

While many improvements have been made, the manuscript still contains numerous deficiencies that warrant rejection, as previously noted. In particular, the discussion section lacks sufficient depth and is overly verbose relative to the experimental data. Major revisions are required.

Major comments

Although the effects of IF and Met are presented side by side, a meaningful comparison is absent. While the authors claim that IF and Met are similar, or IF is superior, this conclusion is premature given the dose-dependent nature of metformin and the limitations of comparing a single dose. Moreover, the underlying mechanisms responsible for the similar effects of IF and metformin remain unexplained. The authors should elucidate whether these effects are attributable to the pharmacological actions of metformin, alterations in the gut microbiota, or other factors.

All results should be moved to the Results section, and redundant information should be removed from the Discussion. In particular, the discussion of changes in mRNA levels for IRS1, IRS2, PI3K.,,, etc. should be consolidated. The authors should provide a clear and concise summary of these changes, supported by relevant literature, to explain why these changes occurred. Only important discussions should be reserved for the Discussion section.

Finally, the authors need to discuss the generalizability of these findings to humans. While this aspect has been a subject of debate in prior research, a further discussion is warranted regarding the extent to which the findings from this experiment can be generalized to human populations. The high-fat, high-carbohydrate diet used in this study may not be representative of the human diet, and the composition of a normal rat diet differs from that of a human diet. Therefore, the authors should consider whether the effects of intermittent fasting observed in rats would be similar in humans, and if so, under what conditions are critical. Additionally, the authors should discuss the potential for obtaining similar results in humans under different intermittent fasting protocols because human diets are inherently intermittent in most cases.

Minor comments

The composition of the diets presented in the Supplementary Table is unclear and does not allow for a meaningful comparison between the standard diet and the HFHC diet. Supplementary Table 1 should be cited where the method HFHC is described (L77).

The abbreviations NPD and PD must be spelled out in the first appearance. Additionally, the group designations NPD, PD, IF, and HFHC-Met are confusing and do not clearly indicate which groups were subjected to a high-fat, high-carbohydrate load. Additionally, the group labels A, B, and C, introduced on lines 102-103 within the Methods section, are absent from the presentation of the results. All group designations should be revised to improve clarity. 
